# End-to-End Goal-Driven Web Navigation

**Rodrigo Nogueira**
Tandon School of Engineering
New York University
rodrigonogueira@nyu.edu

**Kyunghyun Cho**
Courant Institute of Mathematical Sciences
New York University
kyunghyun.cho@nyu.edu

## Abstract

We propose a goal-driven web navigation as a benchmark task for evaluating an agent with abilities to understand natural language and plan on partially observed environments. In this challenging task, an agent navigates through a website, which is represented as a graph consisting of web pages as nodes and hyperlinks as directed edges, to find a web page in which a query appears. The agent is required to have sophisticated high-level reasoning based on natural languages and efficient sequential decision-making capability to succeed. We release a software tool, called WebNav, that automatically transforms a website into this goal-driven web navigation task, and as an example, we make WikiNav, a dataset constructed from the English Wikipedia. We extensively evaluate different variants of neural net based artificial agents on WikiNav and observe that the proposed goal-driven web navigation well reflects the advances in models, making it a suitable benchmark for evaluating future progress. Furthermore, we extend the WikiNav with question-answer pairs from *Jeopardy!* and test the proposed agent based on recurrent neural networks against strong inverted index based search engines. The artificial agents trained on WikiNav outperforms the engined based approaches, demonstrating the capability of the proposed goal-driven navigation as a good proxy for measuring the progress in real-world tasks such as focused crawling and question-answering.

## 1 Introduction

In recent years, there have been many exciting advances in building an artificial agent, which can be trained with one learning algorithm, to solve many relatively large-scale, complicated tasks (see, e.g., [8, 10, 6].) In much of these works, target tasks were computer games such as Atari games [8] and racing car game [6].

These successes have stimulated researchers to apply a similar learning mechanism to *language*-based tasks, such as multi-user dungeon (MUD) games [9, 4]. Instead of visual perception, an agent perceives the state of the world by its written description. A set of actions allowed to the agent is either fixed or dependent on the current state. This type of task can efficiently evaluate the agent's ability of not only in planning but also language understanding.

We, however, notice that these MUD games do not exhibit the complex nature of natural languages to the full extent. For instance, the largest game world tested by Narasimhan et al. [9] uses a vocabulary of only 1340 unique words, and the largest game tested by He et al. [4] uses only 2258 words. Furthermore, the description of a state at each time step is almost always limited to the visual description of the current scene, lacking any use of higher-level concepts present in natural languages.

In this paper, we propose a goal-driven web navigation as a large-scale alternative to the text-based games for evaluating artificial agents with natural language understanding and planning capability. The proposed goal-driven web navigation consists of the whole website as a graph, in which the web pages are nodes and hyperlinks are directed edges. An agent is given a query, which consists of one

or more sentences taken from a randomly selected web page in the graph, and navigates the network, starting from a predefined starting node, to find a target node in which the query appears. Unlike the text-based games, this task utilizes the existing text as it is, resulting in a large vocabulary with a truly natural language description of the state. Furthermore, the task is more challenging as the action space greatly changes with respect to the state in which the agent is.

We release a software tool, called WebNav, that converts a given website into a goal-driven web navigation task. As an example of its use, we provide WikiNav, which was built from English Wikipedia. We design artificial agents based on neural networks (called NeuAgents) trained with supervised learning, and report their respective performances on the benchmark task as well as the performance of human volunteers. We observe that the difficulty of a task generated by WebNav is well controlled by two control parameters; (1) the maximum number of hops from a starting to a target node $N_h$ and (2) the length of query $N_q$.

Furthermore, we extend the WikiNav with an additional set of queries that are constructed from *Jeopardy!* questions, to which we refer by WikiNav-Jeopardy. We evaluate the proposed NeuAgents against the three search-based strategies; (1) SimpleSearch, (2) Apache Lucene and (3) Google Search API. The result in terms of document recall indicates that the NeuAgents outperform those search-based strategies, implying a potential for the proposed task as a good proxy for practical applications such as question-answering and focused crawling.

## 2 Goal-driven Web Navigation

A task $\mathcal{T}$ of goal-driven web navigation is characterized by

$$\mathcal{T} = (\mathcal{A}, s_S, \mathcal{G}, q, R, \Omega). \tag{1}$$

The world in which an agent $\mathcal{A}$ navigates is represented as a graph $\mathcal{G} = (\mathcal{N}, \mathcal{E})$. The graph consists of a set of nodes $\mathcal{N} = \{s_i\}_{i=1}^{N_\mathcal{N}}$ and a set of directed edges $\mathcal{E} = \{e_{i,j}\}$ connecting those nodes. Each node represents a page of the website, which, in turn, is represented by the natural language text $\mathcal{D}(s_i)$ in it. There exists an edge going from a page $s_i$ to $s_j$ if and only if there is a hyperlink in $\mathcal{D}(s_i)$ that points to $s_j$. One of the nodes is designated as a starting node $s_S$ from which any navigation begins. A target node is the one whose natural language description contains a query $q$, and there may be more than one target node.

At each time step, the agent $\mathcal{A}$ *reads* the natural language description $\mathcal{D}(s_t)$ of the current node in which the agent has landed. At no point, the whole world, consisting of the nodes and edges, nor its structure or map (graph structure without any natural language description) is visible to the agent, thus making this task *partially observed*.

Once the agent $\mathcal{A}$ reads the description $\mathcal{D}(s_i)$ of the current node $s_i$, it can take one of the actions available. A set of possible actions is defined as a union of all the outgoing edges $e_{i,\cdot}$ and the *stop* action, thus making the agent have *state-dependent* action space.

Each edge $e_{i,k}$ corresponds to the agent jumping to a next node $s_k$, while the stop action corresponds to the agent declaring that the current node $s_i$ is one of the target nodes. Each edge $e_{i,k}$ is represented by the description of the next node $\mathcal{D}(s_k)$. In other words, deciding which action to take is equivalent to taking a peek at each neighboring node and seeing whether that node is likely to lead ultimately to a target node.

The agent $\mathcal{A}$ receives a reward $R(s_i, q)$ when it chooses the stop action. This task uses a simple binary reward, where

$$R(s_i, q) = \begin{cases} 1, & \text{if } q \subseteq \mathcal{D}(s_i) \\ 0, & \text{otherwise} \end{cases}$$

**Constraints** It is clear that there exists an ultimate policy for the agent to succeed at every trial, which is to traverse the graph breadth-first until the agent finds a node in which the query appears. To avoid this kind of degenerate policies, the task includes a set of four rules/constraints $\Omega$:

1. An agent can follow at most $N_n$ edges at each node.
2. An agent has a finite memory of size smaller than $\mathcal{T}$.

Table 1: Dataset Statistics of WikiNav-4-*, WikiNav-8-*, WikiNav-16-* and WikiNav-Jeopardy.

|  | **WikiNav-4-*** | **WikiNav-8-*** | **WikiNav-16-*** | **WikiNav-Jeopardy** |
|---|---|---|---|---|
| **Train** | 6.0k | 1M | 12M | 113k |
| **Valid** | 1k | 20k | 20k | 10k |
| **Test** | 1k | 20k | 20k | 10k |

3. An agent moves up to $N_h$ hops away from $s_S$.
4. A query of size $N_q$ comes from at least two hops away from the starting node.

The first constraint alone prevents degenerate policies, such as breadth-first search, forcing the agent to make good decisions as possible at each node. The second one further constraints ensure that the agent does not cheat by using earlier trials to reconstruct the whole graph structure (during test time) or to store the entire world in its memory (during training.) The third constraint, which is optional, is there for computational consideration. The fourth constraint is included because the agent is allowed to read the content of a next node.

## 3 WebNav: Software

As a part of this work, we build and release a software tool which turns a website into a goal-driven web navigation task.[1] We call this tool *WebNav*. Given a starting URL, the WebNav reads the whole website, constructs a graph with the web pages in the website as nodes. Each node is assigned a unique identifier $s_i$. The text content of each node $\mathcal{D}(s_i)$ is a cleaned version of the actual HTML content of the corresponding web page. The WebNav turns intra-site hyperlinks into a set of edges $e_{i,j}$.

In addition to transforming a website into a graph $\mathcal{G}$ from Eq. (1), the WebNav automatically selects queries from the nodes' texts and divides them into training, validation, and test sets. We ensure that there is no overlap among three sets by making each target node, from which a query is selected, belongs to only one of them.

Each generated example is defined as a tuple

$$X = (q, s^*, p^*) \tag{2}$$

where $q$ is a query from a web page $s^*$, which was found following a randomly selected path $p^* = (s_S, \ldots, s^*)$. In other words, the WebNav starts from a starting page $s_S$, random-walks the graph for a predefined number of steps ($N_h/2$, in our case), reaches a target node $s^*$ and selects a query $q$ from $\mathcal{D}(s^*)$. A query consists of $N_q$ sentences and is selected among the top-5 candidates in the target node with the highest average TF-IDF, thus discouraging the WebNav from choosing a trivial query.

For the evaluation purpose alone, it is enough to use only a query $q$ itself as an example. However, we include both one target node (among potentially many other target nodes) and one path from the starting node to this target node (again, among many possible connecting paths) so that they can be exploited when training an agent. They are not to be used when evaluating a trained agent.

## 4 WikiNav: A Benchmark Task

With the WebNav, we built a benchmark goal-driven navigation task using Wikipedia as a target website. We used the dump file of the English Wikipedia from September 2015, which consists of more than five million web pages. We built a set of separate tasks with different levels of difficulty by varying the maximum number of allowed hops $N_h \in \{4, 8, 16\}$ and the size of query $N_q \in \{1, 2, 4\}$. We refer to each task by WikiNav-$N_h$-$N_q$.

For each task, we generate training, validation and test examples from the pages half as many hops away from a starting page as the maximum number of hops allowed.[2] We use "Category:Main topic classifications" as a starting node $s_S$.

Table 3: Sample query-answer pairs from WikiNav-Jeopardy.

| Query | Answer |
|---|---|
| For the last 8 years of his life, Galileo was under house arrest for espousing this man's theory. | Copernicus |
| In the winter of 1971-72, a record 1,122 inches of snow fell at Rainier Paradise Ranger Station in this state. | Washington |
| This company's Accutron watch, introduced in 1960, had a guarantee of accuracy to within one minute a month. | Bulova |

As a minimal cleanup procedure, we excluded meta articles whose titles start with "Wikipedia". Any hyperlink that leads to a web page outside Wikipedia is removed in advance together with the following sections: "References", "External Links", "Bibliography" and "Partial Bibliography".

In Table 2, we present basic per-article statistics of the English Wikipedia. It is evident from these statistics that the world of WikiNav-$N_h$-$N_q$ is large and complicated, even after the cleanup procedure.

We ended up with a fairly small dataset for WikiNav-4-*, but large for WikiNav-8-* and WikiNav-16-*. See Table 1 for details.

|  | Hyperlinks | Words |
|---|---|---|
| Avg. | 4.29 | 462.5 |
| $\sqrt{\text{Var}}$ | 13.85 | 990.2 |
| Max | 300 | 132881 |
| Min | 0 | 1 |

Table 2: Per-page statistics of English Wikipedia.

## 4.1 Related Work: Wikispeedia

This work is indeed not the first to notice the possibility of a website, or possibly the whole web, as a world in which intelligent agents explore to achieve a certain goal. One most relevant recent work to ours is perhaps Wikispeedia from [14, 12, 13].

West et al. [14, 12, 13] proposed the following game, called *Wikispeedia*. The game's world is nearly identical to the goal-driven navigation task proposed in this work. More specifically, they converted "Wikipedia for Schools", which contains approximately 4,000 articles as of 2008, into a graph whose nodes are articles and directed edges are hyperlinks. From this graph, a pair of nodes is randomly selected and provided to an agent.

The agent's goal is to start from the first node, navigate the graph and reach the second node. Similarly to the WikiNav, the agent has access to the text content of the current nodes and all the immediate neighboring nodes. One major difference is that the target is given as a whole article, meaning that there is a single target node in the Wikispeedia while there may be multiple target nodes in the proposed WikiNav.

From this description, we see that the goal-driven web navigation is a generalization and re-framing of the Wikispeedia. First, we constrain a query to contain less information, making it much more difficult for an agent to navigate to a target node. Furthermore, a major research question by West and Leskovec [13] was to "*understand how humans navigate and find the information they are looking for*," whereas in this work we are fully focused on proposing an automatic tool to build a challenging goal-driven tasks for designing and evaluating *artificial* intelligent agents.

## 5 WikiNav-Jeopardy: *Jeopardy!* on WikiNav

One of the potential practical applications utilizing the goal-drive navigation is question-answering based on world knowledge. In this Q&A task, a query is a question, and an agent navigates a given information network, e.g., website, to retrieve an answer. In this section, we propose and describe an extension of the WikiNav, in which query-target pairs are constructed from actual *Jeopardy!* question-answer pairs. We refer to this extension of WikiNav by *WikiNav-Jeopardy*.

We first extract all the question-answer pairs from *J! Archive*[3], which has more than 300k such pairs. We keep only those pairs whose answers are titles of Wikipedia articles, leaving us with 133k pairs. We divide those pairs into 113k training, 10k validation, and 10k test examples while carefully

ensuring that no article appears in more than one partition. Additionally, we do not shuffle the original pairs to ensure that the train and test examples are from different episodes.

For each training pair, we find one path from the starting node "Main Topic Classification" to the target node and include it for supervised learning. For reference, the average number of hops to the target node is 5.8, the standard deviation is 1.2, and the maximum and minimum are 2 and 10, respectively. See Table 3 for sample query-answer pairs.

# 6 NeuAgent: Neural Network based Agent

## 6.1 Model Description

**Core Function** The core of the NeuAgent is a parametric function $f_{\text{core}}$ that takes as input the content of the current node $\phi_c(s_i)$ and a query $\phi_q(q)$, and that returns the hidden state of the agent. This parametric function $f_{\text{core}}$ can be implemented either as a feedforward neural network $f_{\text{ff}}$:

$$\mathbf{h}_t = f_{\text{ff}}(\phi_c(s_i), \phi_q(q))$$

which does not take into account the previous hidden state of the agent or as a recurrent neural network $f_{\text{rec}}$:

$$\mathbf{h}_t = f_{\text{rec}}(\mathbf{h}_{t-1}, \phi_c(s_i), \phi_q(q)).$$

We refer to these two types of agents by *NeuAgent-FF* and *NeuAgent-Rec*, respectively. For the NeuAgent-FF, we use a single $\tanh$ layer, while we use long short-term memory (LSTM) units [5], which have recently become *de facto* standard, for the NeuAgent-Rec.

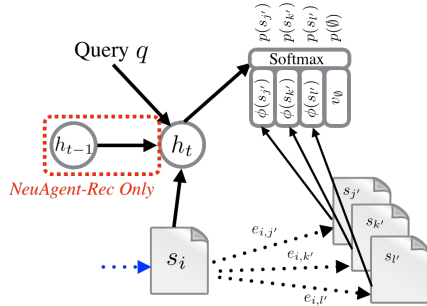

Figure 1: Graphical illustration of a single step performed by the baseline model, NeuAgent.

Based on the new hidden state $\mathbf{h}_t$, the NeuAgent computes the probability distribution over all the outgoing edges $e_i$. The probability of each outgoing edge is proportional to the similarity between the hidden state $\mathbf{h}_t$ such that

$$p(e_{i,j}|\tilde{p}) \propto \exp\left(\phi_c(s_j)^\top \mathbf{h}_t\right). \tag{3}$$

Note that the NeuAgent peeks at the content of the next node $s_j$ by considering its vector representation $\phi_c(s_j)$. In addition to all the outgoing edges, we also allow the agent to *stop* with the probability

$$p(\emptyset|\tilde{p}) \propto \exp\left(\mathbf{v}_\emptyset^\top \mathbf{h}_t\right), \tag{4}$$

where the stop action vector $\mathbf{v}_\emptyset$ is a trainable parameter. In the case of NeuAgent-Rec, all these (unnormalized) probabilities are conditioned on the history $\tilde{p}$ which is a sequence of actions (nodes) selected by the agent so far. We apply a *softmax* normalization on the unnormalized probabilities to obtain the probability distribution over all the possible actions at the current node $s_i$.

The NeuAgent then selects its next action based on this action probability distribution (Eqs. (3) and (4)). If the stop action is chosen, the NeuAgent returns the current node as an answer and receives a reward $R(s_i, q)$, which is one if correct and zero otherwise. If the agent selects one of the outgoing edges, it moves to the selected node and repeats this process of *reading* and *acting*.

See Fig. 1 for a single step of the described NeuAgent.

**Content Representation** The NeuAgent represents the content of a node $s_i$ as a vector $\phi_c(s_i) \in \mathbb{R}^d$. In this work, we use a continuous bag-of-words vector for each document:

$$\phi_c(s_i) = \frac{1}{|\mathcal{D}(s_i)|} \sum_{k=1}^{|\mathcal{D}(s_i)|} \mathbf{e}_k.$$

Each word vector $\mathbf{e}_k$ is from a pretrained continuous bag-of-words model [7]. These word vectors are fixed throughout training.

**Query Representation**   In the case of a query, we consider two types of representation. The first one is a continuous bag-of-words (BoW) vector, just as used for representing the content of a node. The other one is a dynamic representation based on the attention mechanism [2].

In the attention-based query representation, the query is first projected into a set of context vectors. The context vector of the $k$-th query word is

$$\mathbf{c}_k = \sum_{k'=k-u/2}^{k+u/2} \mathbf{W}_{k'}\mathbf{e}_{k'},$$

where $\mathbf{W}_{k'} \in \mathbb{R}^{d \times d}$ and $\mathbf{e}_{k'}$ are respectively a trainable weight matrix and a pretrained word vector. $u$ is the window size. Each context vector is scored at each time step $t$ by $\beta_k^t = f_{\text{att}}(\mathbf{h}_{t-1}, \mathbf{c}_k)$ w.r.t. the previous hidden state of the NeuAgent, and all the scores are normalized to be positive and sum to one, i.e., $\alpha_k^t = \frac{\exp(\beta_k^t)}{\sum_{l=1}^{|q|} \exp(\beta_l^t)}$. These normalized scores are used as the coefficients in computing the weighted-sum of query words to result in a query representation at time $t$:

$$\phi_q(q) = \frac{1}{|q|} \sum_{k=1}^{|q|} \alpha_k^t \mathbf{c}_k.$$

Later, we empirically compare these two query representations.

## 6.2   Inference: Beam Search

Once the NeuAgent is trained, there are a number of approaches to using it for solving the proposed task. The most naive approach is simply to let the agent make a greedy decision at each time step, i.e., following the outgoing edge with the highest probability $\arg\max_k \log p(e_{i,k} | \ldots)$. A better approach is to exploit the fact that the agent is allowed to explore up to $N_n$ outgoing edges per node. We use a simple, forward-only beam search with the beam width capped at $N_n$. The beam search simply keeps the $N_n$ most likely traces, in terms of $\log p(e_{i,k} | \ldots)$, at each time step.

## 6.3   Training: Supervised Learning

In this paper, we investigate supervised learning, where we train the agent to follow an example trace $p^* = (s_S, \ldots, s^*)$ included in the training set at each step (see Eq. (2)). In this case, the cost per training example is

$$C_{\text{sup}} = -\log p(\emptyset | p^*, q) - \sum_{k=1}^{|p^*|} \log p(p_k^* | p_{<k}^*, q). \tag{5}$$

This per-example training cost is fully differentiable with respect to all the parameters of the neural network, and we use stochastic gradient descent (SGD) algorithm to minimize this cost over the whole training set, where the gradients can be computed by backpropagation [11]. This allows the entire model to be trained in an end-to-end fashion, in which the query-to-target performance is optimized directly.

## 7   Human Evaluation

One unique aspect of the proposed task is that it is very difficult for an average person who was not trained specifically for finding information by navigating through an information network. There are a number of reasons behind this difficulty. First, the person must be familiar with, via training, the graph structure of the network, and this often requires many months, if not years, of training. Second, the person must have in-depth knowledge of a broad range of topics in order to make a connection via different concepts between the themes and topics of a query to a target node. Third, each trial requires the person carefully to read the whole content of the nodes as she navigates, which is a time-consuming and exhausting job.

We asked five volunteers to try up to 20 four-sentence-long queries[4] randomly selected from the test sets of WikiNav-$\{4, 8, 16\}$-4 datasets. They were given up to two hours, and they were allowed to

Table 4: The average reward by the NeuAgents and humans on the test sets of WikiNav-$N_h$-$N_q$.

| | $f_{\text{core}}$ | Layers×Units | $\phi_q$ | $N_q = 1$ $N_h = 4$ | 8 | 16 | 2 4 | 8 | 16 | 4 4 | 8 | 16 |
|---|---|---|---|---|---|---|---|---|---|---|---|---|
| (a) | $f_{\text{ff}}$ | $1 \times 512$ | BoW | 21.5 | 4.7 | 1.2 | 40.0 | 9.2 | 1.9 | 45.1 | 12.9 | 2.9 |
| (b) | $f_{\text{rec}}$ | $1 \times 512$ | BoW | 22.0 | 5.1 | 1.7 | 41.1 | 9.2 | 2.1 | 44.8 | 13.3 | 3.6 |
| (c) | $f_{\text{rec}}$ | $8 \times 2048$ | BoW | 17.7 | 10.9 | 8.0 | 35.8 | 19.9 | 13.9 | 39.5 | 28.1 | 21.9 |
| (d) | $f_{\textbf{rec}}$ | $8 \times 2048$ | **Att** | **22.9** | **15.8** | **12.5** | **41.7** | **24.5** | **17.8** | **46.8** | **34.2** | **28.2** |
| (e) | | Humans | | - | - | - | - | - | - | 14.5 | 8.8 | 5.0 |

choose up to the same maximum number of explored edges per node $N_n$ as the NeuAgents (that is, $N_n = 4$), and also were given the option to give up. The average reward was computed as the fraction of correct trials over all the queries presented.

# 8 Results and Analysis

## 8.1 WikiNav

We report in Table 4 the performance of the NeuAgent-FF and NeuAgent-Rec models on the test set of all nine WikiNav-$\{4, 8, 16\}$-$\{1, 2, 4\}$ datasets. In addition to the proposed NeuAgents, we also report the results of the human evaluation.

We clearly observe that the level of difficulty is indeed negatively correlated with the query length $N_q$ but is positively correlated with the maximum number of allowed hops $N_h$. The latter may be considered trivial, as the size of the search space grows exponentially with respect to $N_h$, but the former is not. The former negative correlation confirms that it is indeed easier to solve the task with more information in a query. We conjecture that the agent requires more in-depth understanding of natural languages and planning to overcome the lack of information in the query to find a path toward a target node.

The NeuAgent-FF and NeuAgent-Rec shares similar performance when the maximum number of allowed hops is small ($N_h = 4$), but NeuAgent-Rec ((a) vs. (b)) performs consistently better for higher $N_h$, which indicates that having access to history helps in long-term planning tasks. We also observe that the larger and deeper NeuAgent-Rec ((b) vs (c)) significantly outperforms the smaller one, when a target node is further away from the starting node $s_S$.

The best performing model in (d) used the attention-based query representation, especially as the difficulty of the task increased ($N_q \downarrow$ and $N_h \uparrow$), which supports our claim that the proposed task of goal-driven web navigation is a challenging benchmark for evaluating future progress. In Fig. 2, we present an example of how the attention weights over the query words dynamically evolve as the model navigates toward a target node.

The human participants generally performed worse than the NeuAgents. We attribute this to a number of reasons. First, the NeuAgents are trained specifically on the target domain (Wikipedia), while the human participants have not been. Second, we observed that the volunteers were rapidly exhausted from reading multiple articles in sequence. In other words, we find the proposed benchmark, WebNav, as a good benchmark for machine intelligence but not for comparing it against human intelligence.

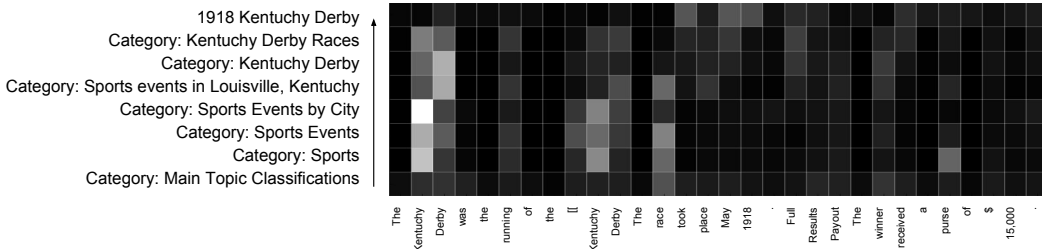

Figure 2: Visualization of the attention weights over a test query. The horizontal axis corresponds to the query words, and the vertical axis to the articles titles visited.

## 8.2 WikiNav-Jeopardy

**Settings** We test the best model from the previous experiment (NeuAgent-Rec with 8 layers of 2048 LSTM units and the attention-based query representation) on the WikiNav-Jeopardy. We evaluate two training strategies. The first strategy is straightforward supervise learning, in which we train a NeuAgent-Rec on WikiNav-Jeopardy from scratch. In the other strategy, we pretrain a NeuAgent-Rec first on the WikiNav-16-4 and finetune it on WikiNav-Jeopardy.

We compare the proposed NeuAgent against three search strategies. The first one, *SimpleSearch*, is a simple inverted index based strategy. SimpleSearch scores each Wikipedia article by the TF-IDF weighted sum of words that co-occur in the articles and a query and returns top-$K$ articles. Second, we use Lucene, a popular open source information retrieval library, in its default configuration on the whole Wikipedia dump. Lastly, we use Google Search API[5], while restricting the domain to wikipedia.org.

Each system is evaluated by document recall at $K$ (Recall@$K$). We vary $K$ to be 1, 4 or 40. In the case of the NeuAgent, we run beam search with width set to $K$ and returns all the $K$ final nodes to compute the document recall. Since there is only one correct document/answer per query, Precision@K = Recall@K / K and therefore we do not show this measure in the results.

Table 5: Recall on WikiNav-Jeopardy. ($\star$) Pretrained on WikiNav-16-4.

| Model | Pre$^\star$ | Recall@1 | Recall@4 | Recall@40 |
|---|---|---|---|---|
| NeuAgent | | 13.9 | 20.2 | 33.2 |
| NeuAgent | ✓ | **18.9** | **23.6** | **38.3** |
| SimpleSearch | | 5.4 | 12.6 | 28.4 |
| Lucene | | 6.3 | 14.7 | 36.3 |
| Google | | 14.0 | 22.1 | 25.9 |

**Result and Analysis** In Table 5, we report the results on WikiNav-Jeopardy. The proposed NeuAgent clearly outperforms all the three search-based strategies, when it was pretrained on the WikiNav-16-4. The superiority of the pretrained NeuAgent is more apparent when the number of candidate documents is constrained to be small, implying that the NeuAgent is able to accurately rank a correct target article. Although the NeuAgent performs comparably to the other search-based strategy even without pretraining, the benefit of pretraining on the much larger WikiNav is clear.

We emphasize that these search-based strategies have access to all the nodes for each input query. The NeuAgent, on the other hand, only observes the nodes as it visits during navigation. This success clearly demonstrates a potential in using the proposed NeuAgent pretrained with a dataset compiled by the proposed WebNav for the task of focused crawling [3, 1], which is an interesting problem on its own, as much of the content available on the Internet is either hidden or dynamically generated [1].

## 9  Conclusion

In this work, we describe a large-scale goal-driven web navigation task and argue that it serves as a useful test bed for evaluating the capabilities of artificial agents on natural language understanding and planning. We release a software tool, called WebNav, that compiles a given website into a goal-driven web navigation task. As an example, we construct WikiNav from Wikipedia using WebNav. We extend WikiNav with *Jeopardy!* questions, thus creating WikiNav-Jeopardy. We evaluate various neural net based agents on WikiNav and WikiNav-Jeopardy. Our results show that more sophisticated agents have better performance, thus supporting our claim that this task is well suited to evaluate future progress in natural language understanding and planning. Furthermore, we show that our agent pretrained on WikiNav outperforms two strong inverted-index based search engines on the WikiNav-Jeopardy. These empirical results support our claim on the usefulness of the proposed task and agents in challenging applications such as focused crawling and question-answering.

## Footnotes

[1] The source code and datasets are publicly available at github.com/nyu-dl/WebNav.

[2] This limit is an artificial limit we chose for computational reasons.

[3] www.j-archive.com

[4] In a preliminary study with other volunteers, we found that, when the queries were shorter than 4, they were not able to solve enough trials for us to have meaningful statistics.

[5] https://cse.google.com/cse

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
