[Reviews · NeurIPS 2016]

Reviewer 1

Summary

This paper introduces a task used as a benchmark in evaluating language understanding agents. The task takes a website and converts it into a graph of individual pages as nodes with hyperlinks represented by edges. Final performance is measured by starting from a page node and reaching a desired target node as represented by a sentence contained within the page for that node. The authors use Wikipedia to instantiate a dataset for this task and evaluate several existing neural net approaches. they also extend this test to include an evaluation of the existing Jeopardy question answer benchmark.

Qualitative Assessment

Overall this is an interesting paper but its contribution is very broad rather than having a specific deeper contribution. the primary contribution seems to be framing Benchmark tests in this web navigation framework however their needs to be more discussion of related work and related benchmark test for comparison when introducing a new benchmark. If this is intended to be the main contribution of the paper the authors could save space by moving discussion of the specific neural network approaches to an appendix. also when introducing this new benchmark it would help to have more in-depth analysis of the dataset and we're evaluating approaches succeed and fail by way of examples in the paper. it's nice that the tool to create this type of data set is being released but I'm skeptical that this framework will create meaningful data sets for anything outside of Wikipedia. again given Wikipedia is the focus more comparison on other evaluations using Wikipedia data are needed. In particular there is some recent work on question answering from Wikipedia tables that seems relevant from the deep learning community. the writing is fairly clear overall but some details are omitted due to packing in a wide variety of information into what could easily be a journal length paper. The wording of constraints 3 and 4 is a bit confusing. It seems like 3 is imposing a maximum number of allowed hops but please clarify. It seems a bit strange to use the number of sentences in a query as the way of defining its difficulty. Sentence lengths can vary widely especially when selected for highly representative sentences of pages. it would be nice to see metrics of query difficulty related to number of words. While it isn't the main focus of this paper a feed forward network with single hidden layer seems obviously too simple for this task. The task as framed isn't a very natural way to explore a resource like Wikipedia so it's not surprising that humans perform poorly. I'm not sure that comparing to human performance is even meaningful other than to say humans aren't good at exploring Wikipedia in this way. It would be nice to see more analysis on the Jeopardy test to understand what about the queries makes the proposed approach outperform standard search engines. intuitively it seems that Jeopardy questions are intentionally difficult when posed purely as search queries.

Confidence in this Review

2-Confident (read it all; understood it all reasonably well)


Reviewer 2

Summary

This paper explores goal-driven web navigation as a task domain for intelligent agents, particularly intelligent agents that have to make use of NLP. The authors formulate a task as navigating from a starting web page to a goal web page, where a goal web page contains a desired query. The authors then provide an empirical study of ANN-based agents trained on navigation dated provided by a tool, WebNav, which can be used to convert websites into the necessary graph format to train such agents.

Qualitative Assessment

This paper is a fun read, and provides insight into web traversal and query answering as a promising benchmark for intelligent agents. The paper has a strong empirical component. In addition to providing a tool to generate data for such problems, the authors train feed forward and recursive neural network based agents and benchmark against traditional information retrieval solutions such as lucene and google, demonstrating improved performance based on precision and recall. The fact that the supporting software and data is released strengthens the paper. While the paper does not make any particularly novel contributions in terms of algorithms, it does provide a foundation for future work in that direction. One question: When comparing against the standard IR search techniques were any steps taken to optimize the query the way a human would? In other words, since the desired target is a sentence from the corpus, a human would quote the sentence in Google rather than allowing it to be tokenized into a bag of words. Not doing this would presumably lead to noisier results. Comment: I agree that applications to targeted web crawling are indeed interesting and should be pursued! Typos: On page 2, the constrains section. The sentence should read "THESE kind of degenerate policies"

Confidence in this Review

2-Confident (read it all; understood it all reasonably well)


Reviewer 3

Summary

The authors propose a benchmark task for evaluating agents capable of some form of natural language understanding: a variation on Wikispeedia introduced by West at al, where an agent read web pages and uses hyperlinks to navigate to connected pages, with the goal of finding a specific keyword or reaching a specific page. The authors provide software to format a website into a dataset usable for this benchmark, and they provide a formatted dataset called WikiNav created by running their software on Wikipedia. The authors introduce 2 neural networks architectures to solve the problem, dubbed "NeuAgents": a recurrent architecture and a feedforward architecture. Results of theses architectures on WikiNav are reported, and a comparison to human performance is provided.

Qualitative Assessment

The paper is interesting and clearly presented. The main flaw would be the lack of originality of the proposed goal-driven web navigation task, since it is almost indistinguishable from the task proposed by West et al and referenced in the present paper. The original contribution of this paper would then be mainly the software and datasets provided, and the experimental results regarding different neural networks architectures on the datasets. It is surprising that the authors feel the need to come up with a new name for their proposed architectures (NeuAgent) since the architectures are not novel nor sophisticated (simple text+query embedding model). It is also a misleading name since these architectures do not fit the usual definition of an "agent" (they are simple feedforward/recurrent classifiers trained in a supervised way). Overall it is a good paper that makes reasonable choices and covers a lot of ground.

Confidence in this Review

2-Confident (read it all; understood it all reasonably well)


Reviewer 4

Summary

In this work, the authors describe a goal-driven web navigation task. Moreover, they release a software tool, named WebNav which compiles a given website into a goal-driven web navigation task.

Qualitative Assessment

Major Comments: 1. It is worth to point out that the proposed task is interesting which the authors try to solve with a neural network based agent via feedforward or recurrent neural network. However, I am wondering is this problem more suitable to be solved with reinforcement learning? To be more specific, with POMDP, where the current states are consist with the hyperlinks of the current page and the final rewards are the hops to the final target. 2. The reward function ( i.e., 2nd formula in page 2) is only a 0/1 loss, why it is not designed corresponding to the hops used to receaching the target page? 3. The organization of the paper is a littele bit confuse. For example, there are four different parts are proposed and similarly named including WebNav, WikiNav, WikiNav-Jeopardy and NeuAgent. Without a strong focus, this paper is a littile bit easy to lost. Minor Comments: 1. The authors only use a wiki dataset to validate the Question - asnwering power of the WikiNav-Jeopardy comparing with Google and Lucene. However, the test pairs from wiki usually have a specific answer (may be only several words), as the examples listed in Table 3 in the paper. It would be interesting to see whether the proposed framework can handle the QA pair whose answer is a paragraph - (e.g., how to install Chrome on the Mac book - 'For the first setp, ... Then,... Finally,'). 2. The aurhtors have used bag-of-words for the content and query representation, it would be interesting to see the performance via using word embedding.

Confidence in this Review

2-Confident (read it all; understood it all reasonably well)


Reviewer 5

Summary

This paper presents a new benchmark task, called goal-driven web navigation, for evaluating an agent's ability to understand natural language and plan on partially observe environment.

Qualitative Assessment

This paper presents a new benchmark task, called goal-driven web navigation, for evaluating an agent's ability to understand natural language and plan on partially observe environment. However, I am not convinced that natural language understanding (NLU) is necessary for such a task. Since a query consists of sentences sampled from a doc, finding the doc given the query is a typical query-doc matching problem. e.g., you can index the doc collection, and use standard IR models to retrieve the doc. The QA task (which is a long-standing task in the research community) is a much better task to evaluate NLU. I am not convinced that the task is a good candidate for evaluating planning on observed environment either. POMDP is well-known in building dialogue systems. The standard dialogue state tracking task, for example, is a good test bed.

Confidence in this Review

2-Confident (read it all; understood it all reasonably well)


Reviewer 6

Summary

This paper presents a software tool, WebNav, to generate a goal-driven web navigation task. A dataset constructed from English Wikipedia is released as benchmark datasets, WikiNav and WikiNav-Jeopardy, for evaluating an agent with capability of processing language and planning the action given a partially observed environment. The paper evaluated different NN-based architectures for the navigation task, and showed that the proposed one achieved better performance compared to traditional IR-based models.

Qualitative Assessment

The provided datasets, WikiNav and WikiNav-Jeopardy, are interesting and can drive more research working on the related task such as reinforcement learning etc. There are some concerns to be addressed: 1) The reward function only considers whether the retuned pages are correct of not. However, for the navigation purpose about arriving the target nodes as soon as possible, in my opinion, including the penalty for taking additional step is needed. For example, the rewards about reaching the target with N steps and with 2*N steps should be different. 2) In the experiments, in addition to showing average reward as the evaluation metric, the average number of steps for navigation (how many nodes being traversed before reaching the target) can be included in Table 4 for better analysis and comparison. 3) It is not clear whether N_n is fixed for all experiments. Is it set to be 4? Why? I would suggest to include more experiments with different N_n, because limiting the number of nodes for exploration can offer richer and more informative analysis of the experiments. 4) The tables should appear with the corresponding texts in the same page. 5) For the evaluation in Table 5, it is not clear to me why the authors used Recall@K instead of Precision@K. To my understanding, IR cares Precision more than Recall. Is there any reason to use Recall for evaluation? I would suggest include both for detailed information. The provided datasets are interesting and useful for the related fields, and addressing the above issues can improve the quality and impact of the paper. If possible, I would suggest the authors release the NeuAgent program for public usage.

Confidence in this Review

2-Confident (read it all; understood it all reasonably well)